# The impact of an eco-score label on US consumers' perceptions of environmental sustainability and intentions to purchase food: A randomized experiment

Lindsey Smith Taillie[1,2]*, Julia A. Wolfson[3,4], Carmen E. Prestemon[1], Maxime Bercholz[1], Laina Ewoldt[2], Phoebe R. Ruggles[5,6], Marissa G. Hall[1,5,6]

1 Carolina Population Center, University of North Carolina at Chapel Hill, Chapel Hill, NC, United States of America, 2 Department of Nutrition, Gillings School of Global Public Health, University of North Carolina at Chapel Hill, Chapel Hill, NC, United States of America, 3 Department of International Health, Johns Hopkins Bloomberg School of Public Health, Baltimore, MD, United States of America, 4 Department of Health Policy and Management, Johns Hopkins Bloomberg School of Public Health, Baltimore, MD, United States of America, 5 Department of Health Behavior, Gillings School of Global Public Health, University of North Carolina at Chapel Hill, Chapel Hill, NC, United States of America, 6 Lineberger Comprehensive Cancer Center, University of North Carolina at Chapel Hill, Chapel Hill, NC, United States of America

* taillie@unc.edu

## Abstract

Front-of-package labels indicating a product's environmental footprint (i.e., eco-score labels) offer promise to shift consumers towards more sustainable food choices. This study aimed to understand whether eco-score labels impacted consumers' perceptions of environmental sustainability and intentions to purchase sustainable and unsustainable foods. US parents (n = 1,013) completed an online experiment in which they were shown 8 food products (4 sustainable and 4 unsustainable). Participants were randomized to a control (n = 503, barcode on product packaging) or eco-score label group (n = 510, eco-score label on product packaging). The eco-score label was color-coded with a grade of A-F based on the product's environmental footprint, where "A" indicates relative sustainability and "F" indicates relative unsustainability. Participants rated each product's environmental sustainability and their future likelihood of purchase. We used multilevel mixed-effects linear regression models and examined moderation by product category and sociodemographic characteristics. The eco-score label lowered perceived sustainability of unsustainable products by 13% in relative terms or -0.4 in absolute terms (95% CI -0.5, -0.3; p<0.001). The eco-score label increased perceived sustainability of sustainable products by 16% in relative terms or 0.6 in absolute terms (95% CI 0.5, 0.7, p<0.001). Effects on purchase intentions were smaller, with a 6% decrease for unsustainable products (p = 0.001) and an 8% increase for sustainable products (p<0.001). For unsustainable products, the effect of eco-score labels on sustainability perceptions was greater for older adults, men, participants with higher educational attainment, and participants with higher incomes. For sustainable products, the effect of ecolabels on sustainability perceptions was greater for those with higher educational attainment. Eco-score labels have the potential to direct consumers

**Data Availability Statement:** The de-identified dataset is publicly available from the Open Science Framework Repository (https://osf.io/vu9t3/).

**Funding:** Initials of funding recipient: M.H Award number: P2CHD050924 Funder website:https://www.nichd.nih.gov/about/org/der/branches/pdb Research reported in this publication was supported by the Eunice Kennedy Shriver National Institute of Child Health & Human Development of the National Institutes of Health under Award Number P2CHD050924. The content is solely the responsibility of the authors and does not necessarily represent the official views of the National Institutes of Health. The funders had no role in study design, data collection and analysis, decision to publish, or preparation of the manuscript.

towards more sustainable products. Future studies should investigate eco-score label effectiveness on behavioral outcomes.

## Introduction

The current food system contributes to an array of environmental harms, including greenhouse gas emissions, resource depletion, and biodiversity loss [1–4]. Many of the foods that are most environmentally damaging are also linked to adverse health outcomes. For example, processed meat is labeled as a Group 1 carcinogen, the same level as tobacco smoke and asbestos [5]. Meat, especially red meat, is also a major driver of greenhouse gas emissions, and the US is a top global consumer [6]. Policies are urgently needed to shift diets towards more sustainable food choices that will both protect the environment and promote public health. For example, a recent paper found that if all consumers who ate high-carbon foods like beef shifted from these higher-carbon foods to lower-carbon foods like chicken, the US could reduce its carbon footprint by over 35% while also improving dietary quality [7].

One policy approach to informing consumers about the environmental impacts of food products in order to improve the sustainability of food choices is to require front-of-package labels on food packages. A wide body of evidence has shown that front-of-package labels that communicate about a food's nutritional properties influence consumers' perceptions of healthfulness and increase their selection of healthier options [8,9]. Given that consumers generally have a low understanding of foods' sustainability [10], front-of-package labels that provide simple, easy-to-understand information on sustainability could facilitate more sustainable choices. As with nutritional labeling [11], environmental labeling could also incentivize the food industry to change production and manufacturing processes, creating more sustainable options.

However, little is known about whether environmental front-of-package labels affect consumers' perceptions of environmental sustainability. In addition, it is unclear whether environmental labels influence consumers' intentions to purchase more- or less- sustainable products. Recent reviews indicate that consumers value environmental labels on food packages [12], are willing to pay more for sustainable products [10], and that environmental labeling is associated with the selection and purchase of more sustainable food products [13]. However, the type of labels included in the reviews were mixed in terms of substance and format (e.g., textual claims about organic, carbon labels with numbers and icons, and certification schemes) reflecting the diverse array of sustainability labels in use across the United States and globally [10]. Thus, it remains unclear which sustainability labels work well to help consumers quickly and easily assess a product's environmental footprint, many of which are not regulated by any third party.

Eco-score labels are one type of environmental label that is of particular interest. Rather than providing numerical information about a product's environmental impacts (e.g., amount of greenhouse gas emissions produced), which consumers have difficulty understanding [14,15], the eco-score label is interpretative. Similar to the Nutri-score label for nutrition labeling, the eco-score label uses an algorithm to assign a score to each product based on its environmental impact, then assigns a grade (A-F) and color to the product to communicate to the consumer whether the product is relatively unsustainable or sustainable. Previous research on the design of environmental labels has shown that use of color coding and grades on environmental labels helps improve effectiveness [15–17]. Data from the UK shows that eco-score labels that combine both colors and grades reduced the environmental impact of products

selected in an online supermarket [18,19], though a similar label had no impact in a worksite cafeteria setting [20].

One additional question is whether eco-score labels will have a similar impact for products that are rated as "sustainable" (i.e., have a relatively low environmental footprint) vs. those that are "unsustainable" (i.e., have a relatively high environmental footprint). In other words, do positive eco-score labels (e.g., a green leaf with an "A" grade) alter consumer perceptions and intentions to purchase as much as negative eco-score labels (e.g., a red leaf with an "F" grade)? There is debate about whether positive labels are as impactful as negative labels. For example, several recent studies have found that negatively framed front-of-package nutrition labels out-performed positive labels on selection of healthier products [21]. In the sustainability labeling literature, a recent UK experiment found that red globes highlighting 'worse' products reduced the environmental impact of shoppers' food selections, but green globes highlighting 'better' products had no effect[18]. A recent US experiment similarly found that negatively framed high climate impact labels outperformed positively framed low climate impact labels at shifting consumers' food choices [22]. However, it is unclear whether similar effects will be observed for eco-score labels.

It is particularly timely and relevant to study this issue among US adults. Some food companies and restaurants are already developing and employing voluntary sustainability labels, though there is little scientific evidence underpinning most of these. In addition, the US Department of Health and Human Services (HHS) and Department of Agriculture (USDA) have launched multiple workgroups on the intersection of climate, nutrition, and health, including one workgroup tasked with assessing the pathways for incorporating environmental sustainability into the US dietary guidelines [23]. Data on consumer responses to environmental labels could provide valuable insight to the food industry and the government on the best strategies for communicating about the sustainability of food choices.

In sum, current research shows that front-of-package nutrition labels promote healthier food choices, but little is known about whether front-of-package environmental labels promote more sustainable food choices. This study investigates the impact of one type of environmental label, the eco-score-label, on consumers' perceptions of sustainability and intentions to purchase foods.

We also aimed to understand differences in responses to eco-score labels that highlight sustainable products vs those that highlight less sustainable products. Specifically, we aimed to examine whether a red eco-score label with an "F" grade would be more impactful at shifting perceptions and intentions than a green eco-score label with an "A" grade. We also explored whether the impact of eco-score labels varied by product category, participant socio-demographic characteristics, and red meat consumption.

## Methods

### Ethics statement

This study was reviewed and approved by the Institutional Review Board at the University of North Carolina, Chapel Hill (#21–3135). All participants provided online written informed consent. The study's design, hypotheses, and analytic plan were pre-registered at https://aspredicted.org/X28_QXJ prior to beginning data collection. De-identified data used in this study are available at Open Science Framework at https://osf.io/vu9t3/.

### Participants

In order to gain an understanding of ecolabels' potential impact in the US, our sample was comprised of US adults. Participants were recruited using Qualtrics Market Research Survey

Panel (www.qualtrics.com), an online survey research panel comprised of a large, diverse sample of US adults that we have previously used to conduct studies on food labeling [24,25]. Inclusion criteria included: residence in the US, age ≥18 years, and being the parent or guardian of any children aged 2 to 12 years or older. This study was conducted as an ancillary study of a trial focused on parents, hence the criteria requiring being a parent or guardian [26]. We recruited a diverse sample, using a quota to ensure that at least 25% of participants self-identified as Hispanic, Latino, or Spanish and at least 25% identified as Black or African American. Qualtrics removed and replaced respondents who did not meet quality control criteria (e.g., completing the entire survey) [26]. The total sample size after quality control measures was 1,013.

## Stimuli

**Product selection and assessment.**    Participants viewed labels on products. In order to facilitate a diverse selection of products with regards to environmental footprint and type of food, we included four categories: sandwiches, pizzas, snacks, and burgers. Within each category, we selected one high- and one low- sustainability product (e.g., veggie and hummus sandwich, ham and cheese sandwich; veggie pizza and pepperoni pizza; almonds and beef jerky; and veggie burger and beef burger). Thus, the study presented participants with a total of 8 products across 4 product categories.

Each product was reviewed by a team of nutritionist research assistants. To determine whether a product was low- or high- sustainability, we used the Food Labeling Toolkit [27], to calculate the environmental footprint for each product. In brief, the toolkit disaggregates meals and snacks into grams of component food groups, then calculates and sums the percent daily value of carbon, nitrogen, and water footprint of each food group to derive an overall score for the product. Since the footprint calculations are based on product weights, we scaled the pairs of products within each food category to have similar weights (for example, both the beef jerky and almonds had weights of 85g). We converted the numerical score into an alphabetical grade ranging from A to F, with A representing the lowest environmental footprint (i.e., most sustainable) and F representing the highest environmental footprint (i.e., least sustainable; we skipped "E", since "F" corresponds to the US academic grading system and is something that consumers are likely to be familiar with). Because we were interested in understanding differential reactions to labels highlighting sustainable vs. unsustainable products, all products in this study received either an A or an F (in each food category, one item of the pair had an A score, and the other item had an F score).

**Labels.**    Fig 1 presents images of a product labeled with study labels. Depending on the study arm, the product displayed one of two types of labels: an eco-score label or a barcode control label. The eco-score labels consisted of color-coded images of leaves with letters A or F with green (A) representing the lowest environmental footprint (i.e. most sustainable) and red (F) representing the highest environmental footprint (i.e. least sustainable). The relevant letter was depicted inside a magnifying glass with the word "ECO-SCORE". The design was based on the Nutri-Score nutrition label [28] and previously tested eco-score labels [18,19]. Similar to previous studies [29,30], a barcode was used as a control label to control for the presence of a label on the front of the package.

**Procedures.**    The experiment was conducted using the Qualtrics survey platform. Data collection took place from December 9, 2022 to December 26, 2022.

After screening for eligibility criteria and providing electronic informed consent, participants completed an online survey. After completing the main experimental task (about prices for sugary drinks) [26], participants were randomly assigned to view one of two labels on

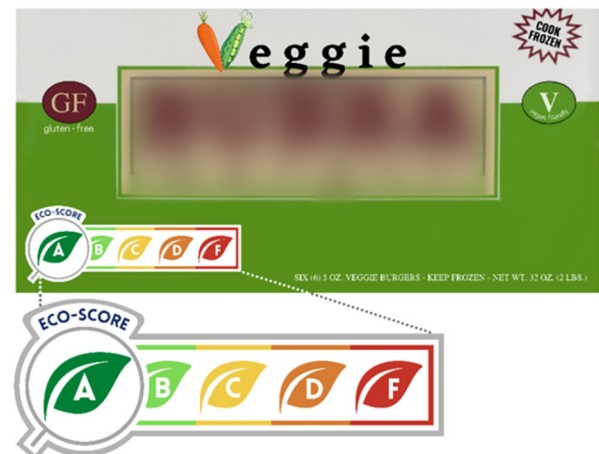

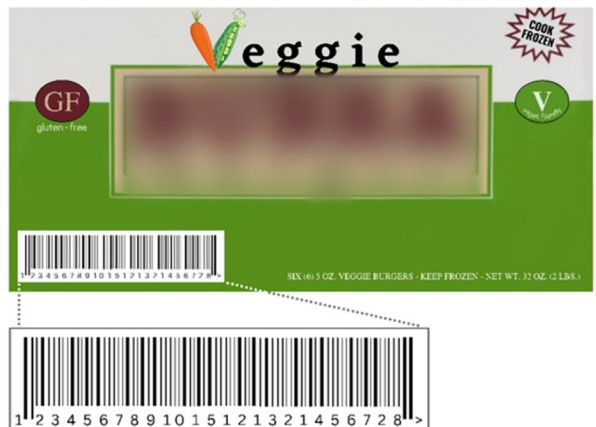

**Fig 1.** Panel A, Veggie Burger with Eco-Score Label; Panel B, Veggie Burger with Control Label*. *Note: Branding has been blurred.

product images: a barcode control label (n = 503) or an eco-score label (n = 510). Randomization was completed between subjects using the Qualtrics randomizer function with a 1:1 allocation ratio. All products in both conditions received a label.

Participants then viewed a set of 8 products in random order and answered questions about each one.

**Measurement.** The primary outcome was perceptions of environmental sustainability of the product. After viewing each product, participants were asked to assess their agreement with the statement, "This product is environmentally friendly." Responses ranged from 1 (strongly disagree) to 5 (strongly agree) [31].

The secondary outcome was intentions to purchase the product. After viewing each product, participants were asked, "How likely would you be to buy this product in the next week, if it were available?" Responses ranged from 1 (not at all likely) to 5 (extremely likely) [32].

Socio-demographic characteristics included age, gender, educational attainment, and race and ethnicity. Because red meat production is a top contributor to dietary greenhouse gas emissions, [33] and the products with low eco-scores all contained red meat, we also assessed red meat consumption. Participants were asked to respond to the question, "In the past 30 days, how often did you eat red meat?" Response options included never, less than 1 time per

week, 1 time per week, 2–3 times per week, 4–6 times per week, 1 time per day, 2 times per day, or 3 or more times per day [24,25,34].

## Statistical analysis

All analyses used Stata (version 18). We used a significance level of 0.05 and statistical tests were two-tailed.

For each outcome, we used a multilevel mixed-effects linear regression model to assess the association between eco-score labeling arm and outcomes, using the following model:

$$y_{ij} = \beta_0 + \beta_1 \times label_i + \beta_2 \times sustainable_j + \beta_3 \times label_i \times sustainable_j + \delta_2 \times pizza_j + \delta_3 \times snack_j + \delta_4 \times burger_j + u_i + v_{ij}$$

where $y_{ij}$ is the outcome for participant i and product j, $label_i$ is a binary variable indicating that participant i is in the eco-score label group (between-subject factor), $sustainable_j$ is a binary variable indicating that product j is a sustainable product carrying a green "A" eco-score label, $pizza_j \ldots burger_j$ are binary variables indicating product j's category (pizza, snack, and burger with respect to sandwich), and $u_i$ and $v_{ij}$ are uncorrelated, normally distributed random effects at the participant and participant-product levels.

Thus, $\beta_1$ is the effect of the eco-score label for unsustainable products carrying a red "F" eco-score label and $\beta_1 + \beta_3$ is the effect of the eco-score label for sustainable products, vis-à-vis the control (barcode). This model allowed us to also test whether eco-score labeling effects differ in magnitude between sustainable vs. unsustainable products.

To explore product category-specific eco-score label effects, we added interactions of the product category with labeling arm ($label_i$), eco-score ($sustainable_j$), and their interaction ($label_i \times sustainable_j$). We then examined the joint statistical significance of the interactions of product category with $label_i$ and $label_i \times sustainable_j$.

In separate models, we also conducted exploratory analyses to examine whether sociodemographic factors (age, gender, educational attainment, and income level) moderated the eco-score label effects on environmental perceptions. We included gender, education, and income as potential moderators due to differences in nutrition label use by these groups [35–37]. We included age as a potential moderator since some research indicates that younger adults are more worried about climate change than older adults [38]. Although it was not pre-registered, we also explored moderation by red meat consumption, since high consumption of red meat could indicate a higher preference for products with low sustainability. For each model, we assessed moderation by including the factor and its interaction with $label_i$, $sustainable_j$, and $label_i \times sustainable_j$.

## Results

The mean age of the sample was 37.1 years (± 9.4) (Table 1). One quarter (25%) of participants identified as Latino, 25% identified as Black or African American, 11% identified as Asian, and 44% of participants identified as white. About a third of participants had a high school degree or less (29%), an associate's degree or some college (36%), and a college degree (36%). About half of the participants had an annual household income of less than $50,000 (46%). About a quarter of the sample ate red meat 1 time per week or less, while 45% ate red meat 2–3 times per week, and about 30% ate it 4–6 times per week or more.

Among unsustainable products, which received the red "F" eco-score label, the eco-score label reduced both the perceived sustainability of products and intentions to purchase products (Table 2, both p<0.01). Among sustainable products, which received the green "A" eco-

**Table 1. Sociodemographic characteristics of the sample.**

| | Control (n = 503) | Ecolabel (n = 510) | Total (n = 1013) |
|---|---|---|---|
| **Age, mean (SD)** | 37.0 (9.0) | 37.1 (9.8) | 37.1 (9.4) |
| **Age Category, n (%)** | | | |
| 18–25 years | 41 (8%) | 51 (10%) | 92 (9%) |
| 26–34 years | 169 (34%) | 170 (33%) | 339 (33%) |
| 35–44 years | 201 (40%) | 191 (37%) | 392 (39%) |
| 45–54 years | 71 (14%) | 66 (13%) | 137 (14%) |
| 55+ years | 21 (4%) | 32 (6%) | 53 (5%) |
| **Gender Identity** | | | |
| Woman | 358 (71%) | 350 (69%) | 708 (70%) |
| Man | 143 (28%) | 157 (31%) | 300 (30%) |
| Non-binary | 2 (0%) | 1 (0%) | 3 (0%) |
| Missing | 0 (0%) | 2 (0%) | 2 (0%) |
| **Race\*** | | | |
| White | 232 (46%) | 212 (42%) | 444 (44%) |
| Hispanic, Latino, or Spanish | 121 (24%) | 130 (25%) | 251 (25%) |
| Black or African American | 120 (24%) | 135 (26%) | 255 (25%) |
| Asian | 49 (10%) | 59 (12%) | 108 (11%) |
| American Indian or Alaska Native | 32 (6%) | 32 (6%) | 64 (6%) |
| Middle Eastern or North African | 4 (1%) | 0 (0%) | 4 (0%) |
| Native Hawaiian or other Pacific Islander | 6 (1%) | 3 (0%) | 9 (1%) |
| Another race or ethnicity | 3 (0%) | 0 (0%) | 3 (0%) |
| Missing | 0 (0%) | 1 (0%) | 1 (0%) |
| **Education level** | | | |
| High school degree/GED or below | 142 (28%) | 150 (29%) | 292 (29%) |
| Associate's degree or some college/technical school | 176 (35%) | 184 (36%) | 360 (36%) |
| Bachelor's degree or higher | 185 (37%) | 176 (35%) | 361 (36%) |
| **Annual household income** | | | |
| $24,999 or less | 106 (21%) | 114 (22%) | 220 (22%) |
| $25,000 to $49,999 | 126 (25%) | 121 (24%) | 247 (24%) |
| $50,000 to $74,999 | 94 (19%) | 108 (21%) | 202 (20%) |
| $75,000 to $99,999 | 76 (15%) | 77 (15%) | 153 (15%) |
| $100,000 or more | 100 (20%) | 90 (18%) | 190 (19%) |
| Missing | 1 (0%) | 0 (0%) | 1 (0%) |
| **Number of people in household, mean (SD)** | 4.0 (1.4) | 4.0 (1.3) | 4.0 (1.4) |
| **Frequency of red meat consumption** | | | |
| 1 time/week or less | 129 (26%) | 127 (25%) | 256 (25%) |
| 2–3 times/week | 233 (46%) | 222 (44%) | 455 (45%) |
| 4–6 times/week | 96 (19%) | 102 (20%) | 198 (20%) |
| 1 or more times/day | 45 (9%) | 59 (12%) | 104 (10%) |

No missing data if missing not reported.

*Response options are not mutually exclusive.

score label, the eco-score label increased both perceived sustainability of products and intentions to purchase products (**Table 2**, both p < .001). The effects were of similar magnitudes for sustainable and unsustainable products in both absolute (Likert scale points) and relative terms. In relative terms, the eco-score label lowered perceived sustainability of unsustainable

**Table 2. Impact of ecolabels on perceptions of environmental sustainability and intentions to purchase products.**

| | Unsustainable products Eco-score F | | Sustainable products Eco-score A | | Magnitude difference between Unsustainable and Sustainable | |
|---|---|---|---|---|---|---|
| | Mean diff.* (95% CI) | p | Mean diff.* (95% CI) | p | Diff.† (95% CI) | p |
| Perceived sustainability | **-0.4** | **<0.001** | **0.6** | **<0.001** | -0.2 | 0.068 |
| | (-0.5, -0.3) | | (0.5, 0.7) | | (-0.3, 0.0) | |
| Purchase intentions | **-0.2** | **0.001** | **0.2** | **<0.001** | 0.0 | 0.721 |
| | (-0.3, -0.1) | | (0.1, 0.4) | | (-0.3, 0.2) | |

Statistically significant results (p < .05) are bolded.

*Difference in means between the eco-label group and the control group.

†Difference between the absolute values of the eco-label's effects for Unsustainable and Sustainable products (i.e. 0.4–0.6 and 0.2–0.2).

products by 13% (from 3.2 to 2.8) and increased that of sustainable products by 16% (from 3.5 to 4.1). Effects on purchase intentions were smaller, however, with a 6% decrease for unsustainable products (from 3.1 to 2.9) and an 8% increase for sustainable products (from 2.9 to 3.1).

There were interactions between the eco-score label and product type (Table 3). For both sustainable and unsustainable products, the eco-score label had the biggest impact on perceived sustainability for sandwiches. For unsustainable products, the eco-score label had the biggest impact on purchasing intentions for pizzas, followed by burgers and sandwiches, whereas there was no statistically significant difference between the control and eco-score arms for snacks. For sustainable products, there was no interaction between the eco-score label and product type on purchasing intentions. Mean unadjusted perceived sustainability and purchase intentions by product type can be found in S1 Table.

The impact of the eco-label (vs control) on perceived sustainability differed by several socio-demographic characteristics, but mostly for unsustainable products (Table 4). For

**Table 3. Impact of ecolabels on perceptions of environmental sustainability and intentions to purchase products, by product category.**

| | Perceived sustainability | | | | Purchase Intentions | | | |
|---|---|---|---|---|---|---|---|---|
| | Unsustainable products Eco-score F | | Sustainable products Eco-score A | | Unsustainable products Eco-score F | | Sustainable products Eco-score A | |
| | Mean diff.* (95% CI) | p | Mean diff.* (95% CI) | p | Mean diff.* (95% CI) | p | Mean diff.* (95% CI) | p |
| Burger | **-0.4** | **<0.001** | **0.5** | **<0.001** | **-0.2** | **0.011** | **0.3** | **0.002** |
| | (-0.5, -0.3) | | (0.3, 0.6) | | (-0.4, -0.0) | | (0.1, 0.4) | |
| Pizza | **-0.3** | **<0.001** | **0.5** | **<0.001** | **-0.3** | **<0.001** | **0.2** | **0.005** |
| | (-0.5, -0.2) | | (0.3, 0.6) | | (-0.5, -0.1) | | (0.1, 0.4) | |
| Sandwich | **-0.6** | **<0.001** | **0.7** | **<0.001** | **-0.2** | **0.004** | **0.2** | **0.033** |
| | (-0.7, -0.4) | | (0.5, 0.8) | | (-0.4, -0.1) | | (0.0, 0.3) | |
| Snack | **-0.3** | **<0.001** | **0.6** | **<0.001** | 0.0 | 0.585 | **0.3** | **<0.001** |
| | (-0.5, -0.2) | | (0.5, 0.8) | | (-0.2, 0.1) | | (0.1, 0.5) | |
| Interaction | | **0.022** | | **0.027** | | **0.032** | | 0.608 |

Statistically significant results (p < .05) are bolded.

*Difference in means between the eco-label group and the control group.

**Table 4. Impact of ecolabels on perceptions of environmental sustainability by age, gender, education, and income.**

| | Unsustainable products Eco-score F | | Sustainable products Eco-score A | |
|---|---|---|---|---|
| | Mean diff.* (95% CI) | *p* | Mean diff.* (95% CI) | *p* |
| **Age**\*\* | | | | |
| 18–25 years | -0.1 | 0.590 | **0.7** | **<0.001** |
| | (-0.4, 0.2) | | (0.4, 1.0) | |
| 26–34 years | **-0.7** | **<0.001** | **0.4** | **<0.001** |
| | (-0.8, -0.5) | | (0.2, 0.6) | |
| 35–44 years | **-0.3** | **0.002** | **0.6** | **<0.001** |
| | (-0.4, -0.1) | | (0.5, 0.8) | |
| 45–54 years | **-0.5** | **<0.001** | **0.7** | **<0.001** |
| | (-0.7, -0.2) | | (0.5, 1.0) | |
| *P for interaction* | | **0.001** | | 0.068 |
| **Gender**\*\* | | | | |
| Man | **-0.6** | **<0.001** | **0.4** | **<0.001** |
| | (-0.8, -0.4) | | (0.3, 0.6) | |
| Woman | **-0.3** | **<0.001** | **0.6** | **<0.001** |
| | (-0.4, -0.2) | | (0.5, 0.7) | |
| *P for interaction* | | **0.018** | | 0.106 |
| **Educational level** | | | | |
| High school/GED or below | -0.2 | 0.061 | **0.4** | **<0.001** |
| | (-0.4, 0.0) | | (0.3, 0.6) | |
| Associate's degree/some college/technical school | **-0.6** | **<0.001** | **0.5** | **<0.001** |
| | (-0.8, -0.4) | | (0.3, 0.7) | |
| Bachelor's degree or higher | **-0.4** | **<0.001** | **0.7** | **<0.001** |
| | (-0.5, -0.2) | | (0.6, 0.9) | |
| *P for interaction* | | **0.003** | | **0.025** |
| **Annual household income** | | | | |
| ≤$24,999 or less | 0.0 | 0.904 | **0.5** | **<0.001** |
| | (-0.2, 0.2) | | (0.3, 0.7) | |
| $25,000- $49,999 | **-0.6** | **<0.001** | **0.5** | **<0.001** |
| | (-0.8, -0.4) | | (0.3, 0.7) | |
| $50,000- $74,999 | **-0.5** | **<0.001** | **0.5** | **<0.001** |
| | (-0.7, -0.3) | | (0.2, 0.7) | |
| $75,000-$99,999 | **-0.3** | **0.021** | **0.9** | **<0.001** |
| | (-0.5, -0.0) | | (0.6, 1.1) | |
| ≥$100,000 | **-0.6** | **<0.001** | **0.7** | **<0.001** |
| | (-0.8, -0.4) | | (0.5, 0.9) | |
| *P for interaction* | | **<0.001** | | 0.060 |
| **Red meat consumption** | | | | |
| 1 time/week or less | -0.4 | **<0.001** | 0.7 | **<0.001** |
| | (-0.6, -0.2) | | (0.5, 0.9) | |
| 2–3 times/week | -0.5 | **<0.001** | 0.6 | **<0.001** |
| | (-0.7, -0.4) | | (0.4, 0.7) | |
| 4–6 times/week or more | -0.3 | **0.004** | 0.5 | **<0.001** |
| | (-0.4, -0.1) | | (0.3, 0.6) | |

*(Continued)*

**Table 4.** (Continued)

| | Unsustainable products Eco-score F | | Sustainable products Eco-score A | |
|---|---|---|---|---|
| | Mean diff.* (95% CI) | *p* | Mean diff.* (95% CI) | *p* |
| *P for interaction* | | 0.075 | | 0.271 |

Statistically significant results (*p* < .05) are bolded.

*Difference in means between the eco-label group and the control group.

** Age = 55+ years and gender = non-binary excluded due to small numbers of respondents per arm.

unsustainable products, eco-score labels reduced perceptions of sustainability for adults over age 25 years, but not for adults aged 18–25 years (p for interaction = 0.001). For unsustainable products, eco-score labels had a bigger impact on reducing perceived sustainability for men compared to women (p for interaction = 0.018). They also had a bigger impact on reducing perceived sustainability for people with more than a high school degree (p for interaction = 0.003), with the biggest effects observed among those with an associate's degree/some college (-0.6, 95% CI -0.8, -0.4). Eco-score labels had a bigger impact among those with income above $25,000 (p for interaction< 0.001), with the largest effects observed among people with incomes of $25,000 to $49,0000 or over $100,000. For sustainable products, the only interaction observed was for education; eco-score labels increased perceptions of sustainability more for participants with higher vs. lower educational attainment (p for interaction = 0.025). There were no differences in eco-score label impact by red meat consumption for either sustainable or unsustainable products.

## Discussion

In this online experiment of US consumers, we found that eco-score labels impacted consumers' perceptions of sustainability and intentions to purchase products, relative to a control label. As predicted, eco-score labels increased sustainability perceptions and intentions to purchase sustainable products that received an "A" grade for environmental footprint, whereas the labels decreased sustainability perceptions and intentions to purchase unsustainable products that received an "F" grade for environmental footprint. Though more research is needed, these results show promise that an eco-score front of package label has the potential to shift consumer perceptions and purchases to promote more sustainable food choices.

In this study, the magnitude of the effect of the eco-label on perceptions of sustainability was similar for sustainable and unsustainable products. These results contrasted with our expectations, which was that there would be a larger impact for unsustainable products as consumers would respond more to negative information on the eco-score label (i.e., red color, F grade) than positive information (i.e., green color, A grade). Our results also are in contrast to the aforementioned findings from a UK experiment on eco-score labels, which found that red globes highlighting unsustainable products, but not green globes highlighting sustainable products, reduced the environmental impact of consumer selections in an online supermarket [18]. The difference between studies could be because our eco-score labels did not contain a single red or green icon but rather the entire Eco-score, with the relevant grade magnified. Indeed, in the same UK study, researchers found that eco-score labels that depicted the spectrum of eco-scores, like ours, led to reductions in the environmental footprint of selections. One possibility is that for sustainable products, a labeling system that shows the spectrum of eco-scores is more effective than a label that depicts only a single positive icon. Future research

should explore how different label designs and scoring systems impact consumer perceptions and behaviors.

We found evidence to suggest that eco-score labels may have more impact in some product categories than others. For example, among unsustainable products, the eco-score label had little or no effects on changing sustainability perceptions or intentions to purchase snacks. For both sustainable and unsustainable products, the eco-score label had the biggest impact on sustainability perceptions for sandwiches. Previous research on nutrition labeling suggests that labels are most impactful for product categories where consumers have less understanding and knowledge [39,40]. Consumers may have had less prior knowledge about the environmental footprint of sandwiches, leading to a bigger impact for this category. However, we cannot rule out that these results are due to the types of products we chose. For example, the unsustainable snack was beef jerky; it is possible that consumers were already aware that beef has a high environmental impact and thus the inclusion of the eco-label did not affect their perceptions or intentions. Future research including a broader range of products with more diverse eco-scores will be necessary to understand whether eco-scores work similarly across product categories.

We found moderation of eco-score labels by socio-demographic characteristics, mostly for unsustainable products. Eco-score labels had more impact on sustainability perceptions for older adults vs. younger adults and for men vs. women. Previous literature indicates that younger adults as well as women have a higher awareness for and concern for the environmental sustainability of foods [41,42]. In this study, one possibility eco-score labels were less effective at shifting perceptions among these groups is because baseline levels of awareness were already high.

Other patterns of moderation were less clear. For both sustainable and unsustainable products, we observed a larger effect of eco-labels for participants with higher education vs. those with a high school degree or less. There was a similar pattern based on household income, with a larger effect of eco-labels among participants with higher incomes compared to the lowest income category; however, there was not a stepwise increase in impact with increasing income. One possibility is participants of lower education or income understood the eco-labels less well, minimizing the impact of the eco-labels on sustainability perceptions. Recent reviews of sustainability labels on foods have found inconsistent results regarding the role of sociodemographic characteristics [10,13], suggesting more research in this space is needed.

Our results have implications for US and global policy discussions on environmental sustainability labeling. In the US, current efforts remain focused on voluntary labels, typically including a positive icon or badge for more sustainable products. For example, in 2020, Panera, a large fast-casual restaurant chain, made headlines for its "Cool Food Meal" label, which indicated items with a lower carbon footprint [43]. From a regulatory perspective, although no countries currently require the inclusion of environmental sustainability labels on foods [44], France is developing an eco-score label for foods, with an evaluation anticipated in 2024 and a mandatory roll-out in 2025 (correspondence with ADEME, France's ecological transition agency) [45]. In the UK, politicians and environmental organizations have also called for the creation of a systematic environmental label on foods [46,47]. These voluntary and regulatory initiatives offer an opportunity to go beyond the present research on sustainability perceptions to understand whether and how eco-score labels actually influence consumer purchases [46,48].

In addition, an important area for future research will be understanding the nutritional implications of sustainability labeling. A recent experimental study in Germany that tested both a Nutri-score Label and an eco-Score label found that the labels influenced each other's perceived healthfulness and sustainability, potentially resulting in inaccurate perceptions [49].

For example, French fries that displayed both an eco-score label and a Nutri-score label were perceived as healthier than fries that displayed only a Nutri-score label. The results also raised the possibility that the use of nutrition labels and environmental labels together could cause confusion or generate internal conflict for consumers when the labels indicate dissonant values (e.g., for high on nutrition but low on sustainability, or vice versa). Future research in US populations would be useful to understand how a joint nutrition- and- environmental labeling system would work among American consumers.

Strengths of the study include recruitment of a diverse sample with respect to race/ethnicity and the experimental design using multiple product types. This study also had several limitations. One limitation was that the products in our study carried only "A" and "F" labels. Future research should explore how eco-score labels affect perceptions and intentions across a more diverse range of products that include a variety of eco-scores. Similarly, our study asked participants to consider products for a brief time in an online setting, whereas real-world food purchasing settings are much more complex. Future research should include both a broader population of consumers as well as a more realistic food purchasing environment to understand whether and how eco-score labels impact actual food selection behaviors.

## Conclusions

The objective of this study was to examine the impact of eco-score labels on consumers' perceptions of sustainability and intentions to purchase foods. In a randomized experiment of US adults, relative to a control label, eco-score labels changed consumers' perceptions about the sustainability of foods and their intentions to purchase foods, with some differences by food category and participant characteristics. This preliminary evidence suggests that eco-labels could be a useful policy tool or voluntary approach to shift consumers towards more sustainable food choices; results are timely and relevant for voluntary and governmental sustainability labeling initiatives in the US and globally. However, a key limitation of this study was that it used an artificial online setting to measure labels' impact on food purchasing intentions rather than on actual food purchases. Future research is necessary to test the impact of eco-score labels on objective behavioral outcomes in more realistic food retail environments.

## Supporting information

**S1 Table. Mean (SD) perceived sustainability and purchase intentions by product category and score type.**
(DOCX)

## Acknowledgments

The authors would like to thank Sara Cathey for the design of the eco-score labels and Veronica Lippuner for project management support.

## Author Contributions

**Conceptualization:** Lindsey Smith Taillie.

**Formal analysis:** Maxime Bercholz.

**Funding acquisition:** Lindsey Smith Taillie, Marissa G. Hall.

**Investigation:** Carmen E. Prestemon.

**Methodology:** Lindsey Smith Taillie, Maxime Bercholz.

**Project administration:** Carmen E. Prestemon.

**Supervision:** Lindsey Smith Taillie, Marissa G. Hall.

**Writing – original draft:** Lindsey Smith Taillie, Maxime Bercholz.

**Writing – review & editing:** Julia A. Wolfson, Carmen E. Prestemon, Laina Ewoldt, Phoebe R. Ruggles, Marissa G. Hall.

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
