## [Decision Letter · Decision Letter 0]

9 May 2024

PONE-D-24-04592The impact of an eco-score label on US consumers’ perceptions of environmental sustainability and intentions to purchase food: A randomized experimentPLOS ONE

Dear Dr. Taillie,

Thank you for submitting your manuscript to PLOS ONE. After careful consideration, we feel that it has merit but does not fully meet PLOS ONE’s publication criteria as it currently stands. Therefore, we invite you to submit a revised version of the manuscript that addresses the points raised during the review process.

**The paper should be restructured by highlighting problem statement, contribution and significance of the study and justify the methodology. **

We look forward to receiving your revised manuscript.

Kind regards,

Abid Rashid Gill

Academic Editor

PLOS ONE

Journal Requirements:

Reviewers' comments:

Reviewer's Responses to Questions

**Comments to the Author**

1. Is the manuscript technically sound, and do the data support the conclusions?

Reviewer #1: Yes

Reviewer #2: Yes

2. Has the statistical analysis been performed appropriately and rigorously? 

Reviewer #1: Yes

Reviewer #2: Yes

3. Have the authors made all data underlying the findings in their manuscript fully available?

Reviewer #1: Yes

Reviewer #2: Yes

4. Is the manuscript presented in an intelligible fashion and written in standard English?

Reviewer #1: Yes

Reviewer #2: Yes

5. Review Comments to the Author

**Reviewer #1**: Good work, the overall article is very well written and analysed

**Reviewer #2:** problem statement need to be more rigor and research gaps should be justified. please mention justification to consider the current population and sample of study. Need justification to use Qualtrics Market Research Survey Panel. Statistical analysis seem ok. While discussion need improvement. Conclusion is very limited need to be comprehensive and precise.

6. PLOS authors have the option to publish the peer review history of their article (what does this mean?). If published, this will include your full peer review and any attached files.

Reviewer #1: **Yes: **Saghir Pervaiz Ghauri

Reviewer #2: No

---

## [Author Response · Author response to Decision Letter 0]

5 Jun 2024

Thank you for your review and comments. We have included specific responses to your comments in the attached "Response to Reviewers" file. We believe that these revisions have resulted in a stronger presentation of our work.

---

## [Editor Report · Decision Letter 1]

12 Jun 2024

The impact of an eco-score label on US consumers’ perceptions of environmental sustainability and intentions to purchase food: A randomized experiment

PONE-D-24-04592R1

Dear authors, 

We’re pleased to inform you that your manuscript has been judged scientifically suitable for publication and will be formally accepted for publication once it meets all outstanding technical requirements.

Kind regards,

Abid Rashid Gill

Academic Editor

PLOS ONE
---

## [Editor Report · Acceptance letter]

18 Jun 2024

PONE-D-24-04592R1 

PLOS ONE

Dear Dr. Taillie, 

I'm pleased to inform you that your manuscript has been deemed suitable for publication in PLOS ONE. Congratulations! Your manuscript is now being handed over to our production team.

Kind regards, 

on behalf of

Dr. Abid Rashid Gill 

Academic Editor

PLOS ONE